# New Multitarget Rivastigmine–Indole Hybrids as Potential Drug Candidates for Alzheimer’s Disease

**DOI:** 10.3390/pharmaceutics16020281

**Published:** 2024-02-16

**Authors:** Leo Bon, Angelika Banaś, Inês Dias, Inês Melo-Marques, Sandra M. Cardoso, Sílvia Chaves, M. Amélia Santos

**Affiliations:** 1Centro de Química Estrutural, Institute of Molecular Sciences, Departamento de Engenharia Química, Instituto Superior Técnico, Universidade de Lisboa, Av. Rovisco Pais 1, 1049-001 Lisboa, Portugal; leo@pavaleo.fr (L.B.); angeban429@student.polsl.pl (A.B.); ines.ferreira.dias@tecnico.ulisboa.pt (I.D.); 2CNC-Center for Neuroscience and Cell Biology, University of Coimbra, 3004-504 Coimbra, Portugal; ines.marques@cnc.uc.pt (I.M.-M.); sicardoso@fmed.uc.pt (S.M.C.); 3Faculty of Medicine, University of Coimbra, 3004-504 Coimbra, Portugal

**Keywords:** Alzheimer´s disease, multitarget drugs, rivastigmine–indole, antioxidants, cholinesterases, amyloid-β aggregation, neuroprotection

## Abstract

Alzheimer’s disease (AD) is the most common form of dementia with no cure so far, probably due to the complexity of this multifactorial disease with diverse processes associated with its origin and progress. Several neuropathological hallmarks have been identified that encourage the search for new multitarget drugs. Therefore, following a multitarget approach, nine rivastigmine–indole (RIV-IND) hybrids (**5a1-3**, **5b1-3**, **5c1-3**) were designed, synthesized and evaluated for their multiple biological properties and free radical scavenging activity, as potential multitarget anti-AD drugs. The molecular docking studies of these hybrids on the active center of acetylcholinesterase (AChE) and butyrylcholinesterase (BChE) suggest their capacity to act as dual enzyme inhibitors with probable greater disease-modifying impact relative to AChE-selective FDA-approved drugs. Compounds **5a3** (IC_50_ = 10.9 µM) and **5c3** (IC_50_ = 26.8 µM) revealed higher AChE inhibition than the parent RIV drug. Radical scavenging assays demonstrated that all the hybrids containing a hydroxyl substituent in the IND moiety (**5a2-3**, **5b2-3**, **5c2-3**) have good antioxidant activity (EC_50_ 7.8–20.7 µM). The most effective inhibitors of Aβ_42_ self-aggregation are **5a3**, **5b3** and **5c3** (47.8–55.5%), and compounds **5b2** and **5c2** can prevent the toxicity induced by Aβ_1-42_ to cells. The in silico evaluation of the drug-likeness of the hybrids also showed that all the compounds seem to have potential oral availability. Overall, within this class of RIV-IND hybrids, **5a3** and **5c3** appear as lead compounds for anti-AD drug candidates, deserving further investigation.

## 1. Introduction

Alzheimer´s disease (AD) is the most common age-depending neurodegenerative disorder, leading to severe dementia for which no curative treatment options are available. Nowadays, it is affecting 50 million elderly people worldwide, but the number of patients is expected to rise continuously and reach about 130 million by 2050, following the trend of life expectancy increase [1,2]. Although AD etiology and pathogenesis remain elusive, significant atrophy of the cortex and hippocampus can be detected, while several neuropathological hallmarks have already been identified. The AD patient´s brain is characterized by misaccumulation of proteins, namely the extracellular β-amyloid (Aβ) plaques and the intracellular hyperphosphorylated tau protein aggregates in neurofibrillary tangles (NFT), as well as a great deficit on the cholinergic neurotransmitters [3,4,5]. However, other associated etiological hypotheses have been gaining attention, such as the dyshomeostasis of biometals, elevated levels of reactive oxygen species (ROS) and oxidative stress [6,7].

Despite intensive research efforts on AD, until now, the FDA (Food and Drug Administration, USA) has approved treatments mostly based on cholinergic drugs to compensate for the loss of cholinergic neurons, namely acetylcholinesterase (AChE) and butyrylcholinesterase (BChE) inhibitors, such as tacrine, donepezil, rivastigmine, galantamine, but also one glutamatergic drug (memantine), a *N*-methyl-*D*-aspartate (NMDA) receptor antagonist [8]. Quite recently, a new generation of monoclonal antibodies (aducanumab, lecanemab, donanemab) [9,10,11], directed to slow down the formation of Aβ aggregates [12], have also been approved by the FDA, although they have not yet been approved by the EMA (European Medicinal Agency) due to adverse effects and also questionable efficacy [13]. Unfortunately, the approved drugs only provide symptomatic relief for mild forms of AD, but they do not cure or slow down the progression of the disease [14].

Therefore, at present, no disease-modifying drugs are available for the treatment of AD. The inefficacy of current therapies has been attributed to the complex multifactorial nature of AD, involving several pathophysiological processes, some of them interconnected. In view of this, there is a current strategy for drug development, based on multitarget directed ligands (MTDL), to enable the simultaneous hitting of several of the most important targets associated with AD instead of pursuing the current single-target drugs. Hence, medicinal chemists have been working to design single drugs with multiple pharmacophores that can act on the various targets of AD, including cholinergic and amyloidogenic processes, oxidative stress, neuroinflammation, metal chelation and also inhibition of other enzymes with significant roles in these neurodegenerative processes [15,16,17,18,19].

Indole is a privileged scaffold that can gain versatile biological activities and pharmacophoric properties. It is found in several synthetic and semisynthetic therapeutic agents [20], and it is also the core nucleus of diverse naturally occurring compounds, namely those found in bioactive endogenous substances (e.g., melatonin, serotonin, tryptophan), in plants and in alkaloids with demonstrated neuroprotective actions in neurodegenerative diseases such as AD [21]. Therefore, based on the recognized neuroprotective roles of indole-containing molecules, such as melatonin, which drops down in AD patients [22], several medicinal chemistry researchers have recently combined the indole moiety with various bioactive moieties (e.g., 8-hydroxy-quinolines, tacrine) to develop pleiotropic compounds aimed to treat AD, as reported in a recent review [23].

Following this strategy in our current approach for MTDLs, in which single small molecules enclose the active moiety of the drug rivastigmine (RIV) [24], we propose to develop rivastigmine–indole (RIV-IND) hybrids. In particular, a RIV unit is coupled with an indole (IND) moiety (as a surrogate of melatonin), which is conveniently substituted with a hydroxyl group and linked to the RIV moiety to assure the inhibition of both cholinesterases (AChE, BChE) and also other multiple important capacities, such as inhibition of Aβ aggregation, antioxidant activity (ROS scavenging) and biometal chelation. Therefore, pursuing our design strategy, it is possible to assemble all these properties in a small molecule with only two active moieties (bis-hybrid) instead of the reported large tris-hybrid indole derivatives, which also include both tacrine and ferulic acid moieties [25]. Herein, we report the study of a set of nine novel rivastigmine–indole hybrids (see Figure 1), including the design, development and evaluation of their biological activities (inhibition of AChE and BChE; inhibition of Aβ aggregation (in the presence and absence of Cu^2+^); in vitro neuroprotection in SH-SY5Y cells subjected to neuro-stressors) and antioxidant ability (radical scavenging). Structure–activity relationships are rationalized considering the differences in the indole substituents and the size of the linker between both main moieties.

## 2. Materials and Methods

### 2.1. Materials and Equipment

Reagents of analytical grade were used, and solvents were dried according to standard methods [26]. TLC was performed to monitor chemical reactions using aluminum plates coated with silica gel 60 F_254_ from Merck (Darmstadt, Germany). Column chromatographic separations were performed using silica gel 60A 70–200 µ from Carlo Erba Reagents. Melting points (M.P.) were measured using a Leica Galen III hot-stage microscope. The NMR spectra (^1^H and ^13^C) were recorded on a Bruker AVANCE III-400 NMR spectrometer (at 400 and 100 MHz). Tetramethylsilane (TMS) was used as a standard internal reference to report the chemical shifts (δ). Abbreviations: s = singlet, d = doublet, t = triplet, q = quartet, m = multiplet, bs = broad singlet. Mass spectra (ESI-MS) were obtained on a 500 MS LC Ion Trap mass spectrometer (Varian Inc., Palo Alto, CA, USA) equipped with an ESI ion source, operated in the positive ion mode. High-resolution mass spectra (HRMS) were obtained on a Bruker Impact II quadrupole mass spectrometer (Bruker Daltoniks, Billerica, MA, USA). UV-Vis spectrophotometers were used: Perkin Elmer Lambda 35 (radical scavenging, cholinesterase inhibition) and Spectramax Plus 384 (cell assays). Fluorescence measurements were performed on a microplate reader, BMG Labtech, POLARstar OPTIMA, for evaluation of Aβ_42_ aggregation.

### 2.2. Synthesis of the RIV-IND Hybrids

The synthetic steps involved in the preparation of the rivastigmine–indole (RIV-IND) hybrids and their intermediates are schematically represented in Figure 2.

#### 2.2.1. General Procedure for the Synthesis of the Carbamates (**2a**, **2b**, **2c**)

To a solution of 1.62 mmol of the phenolic derivative (3-nitrophenol, 3-cyanophenol, 2-(3-hydroxyphenyl)acetonitrile) in 1 mL of triethylamine (TEA), 1.65 mmol of *N*-ethylmethylcarbamoyl chloride was added. The mixture was stirred at 95 °C for 12 h. Afterward, the reaction mixture was added to CH_2_Cl_2_, which was washed with NaOH 1M aqueous solution. The organic phase was dried over anhydrous Na_2_SO_4_, filtrated, roto-evaporated and then dried under vacuum to obtain the desired oiled compound.

##### 3-Nitrophenyl ethylmethylcarbamate (**2a**)

The title compound (**2a**) was obtained from 3-nitrophenol as a pale brown oil. The completion of the reaction was controlled by TLC (CH_2_Cl_2_/MeOH, 60/1). Yield = 82.4%. ^1^H NMR (400 MHz, MeOD-*d*_4_), δ (ppm): 8.08 (d,1H, *J* = 8 Hz, *Ph-CH*), 8.02 (1H, s, *Ph-CH*), 7.62 (dd, 1H, *J* = 8 Hz, *Ph-CH*), 7.53 (d, 1H, *J* = 8 Hz, *Ph-CH*); 3.52, 3.42 (2 × q, 2H, *J* = 6 Hz, N*CH*_2_CH_3_ rotamers); 3.12, 3.00 (2 × s, 3H, N*CH*_3_ rotamers); 1.27, 1.20 (2 × t, 3H, *J* = 6 Hz, NCH_2_*CH*_3_ rotamers). MS-ESI (*m*/*z*): 225.06 (M + 1)^+^.

##### 3-Cyanophenyl ethylmethylcarbamate (**2b**)

The title compound (**2b**) was obtained from 3-cyanophenol, affording a brown pale oil. The completion of the reaction was controlled by TLC (CH_2_Cl_2_/MeOH, 50/1). Yield = 92.3%. ^1^H NMR (400 MHz, MeOD-*d*_4_), δ (ppm): 7.60–7.54 (m, 3H, Ph*-CH*) 7.45 (d, 1H, *J* = 8 Hz, Ph*-CH*), 3.51, 3.40 (2 × q, *J* = 6 Hz, N*CH*_2_CH_3_ rotamers), 3.10, 2.99 (2 × s, 3H, N*CH*_3_ rotamer), 1.26, 1.19 (2 × t, 3H, *J* = 8 Hz, NCH_2_*CH*_3_ rotamers). MS-ESI (*m*/*z*): 205.10 (M + 1)^+^.

##### 3-(Cyanomethyl)phenyl ethylmethylcarbamate (**2c**)

The title compound (**2c**) was obtained from (3-hydroxyphenyl)acetonitrile as a pale brown oil. The completion of the reaction was controlled by TLC (CH_2_Cl_2_/MeOH, 30/1). Yield = 75.6%. ^1^H NMR (400 MHz, MeOD-*d*_4_), δ (ppm): 7.42 (t, 1H, *J* = 8 Hz, *Ph-CH*), 7.24 (d, 1H, *J* = 8 Hz, *Ph-CH*), 7.15 (s, 1H, *Ph*), 7.11 (d, 1H, *J* = 8 Hz, *Ph-CH*), 3.92 (s, 2H, Ph*CH*_2_CN); 3.52, 3.41 (2 × q, *J* = 7 Hz, N*CH*_2_CH_3_ rotamers); 3.11, 3.10 (2 × s, 3H, N*CH*_3_ rotamer); 1.28, 1.20 (2 × t, 3H, *J* = 8 Hz, NCH_2_*CH*_3_ rotamers). MS-ESI (*m*/*z*): 219.15 (M + 1)^+^.

#### 2.2.2. General Procedure for the Synthesis of the Amino-phenylcarbamates (**3a**, **3b**, **3c**)

The nitro- or cyano-carbamate derivatives (**2a-c**) (3-nitrophenyl ethylmethylcarbamate, 3-cyanophenyl ethylmethylcarbamate, 3-(cyanomethyl) phenyl ethylmethylcarbamate) were subjected to hydrogenolysis. The compounds (1.40–1.50 mmol) were dissolved in MeOH (15–20 mL) and then 2.12 mmol of 10% Pd-C was added. This suspension was put under a hydrogen atmosphere (4 bars) and shaken for 4 h at r.t. Afterward, the catalyst was filtered off, and the solution was evaporated until dry, providing the final amine as an oil of compounds **3a** and **3b**. For the hydrogenolysis of compound **3c**, an identical procedure was followed, but some drops of concentrated HCl were added to the methanolic suspension. Then, the catalyst was filtered off, and the solution was evaporated to dryness. The residue was diluted with CH_2_Cl_2_ and washed with NaOH 1 M and H_2_O until pH > 7. The organic layer phase was dried over anhydrous Na_2_SO_4_ and roto-evaporated, and the oily residue was dried under vacuum. The crude oil compound was purified by chromatography column.

##### 3-Aminophenyl ethylmethylcarbamate (**3a**)

The title compound was synthesized from 3-nitrophenyl ethylmethylcarbamate (**2a**), and it was obtained as a pale yellow oil. The completion of the reaction was controlled by TLC (CH_2_Cl_2_/MeOH, 60/1). Yield = 46.9%. ^1^H NMR (400 MHz, MeOD-*d*_4_), δ (ppm): 7.05 (t, 1H, *J* = 9 Hz, *Ph-CH*), 6.54 (d, 1H, *J* = 8 Hz, *Ph-CH*), 6.42 (s, 1H, *Ph-CH*), 6.36 (d, 1H, *J* = 8 Hz, *Ph-CH*); 3.47, 3.37 (2 × q, *J* = 8 Hz, N*CH*_2_CH_3_ rotamers); 3.06, 2.95 (2 × s, 3H, N*CH*_3_ rotamers); 1.24, 1.16 (2 × t, 3H, *J* = 8 Hz, CH_2_*CH*_3_ rotamers). MS-ESI (*m*/*z*): 195.05 (M + 1)^+^.

##### 3-(Aminomethyl)phenyl ethylmethylcarbamate (**3b**)

This compound was synthesized from 3-cyanophenyl ethylmethylcarbamate (**2b**). The completion of the reaction was controlled by TLC (EtOAc/NH_3_, 98/2). The crude oil of the compound was further purified by chromatography column (eluent EtOAc/NH_3_, 98/2), affording the title pure compound (**3b**) as a pale-yellow oil. Yield = 70.7%. ^1^H NMR (400 MHz, MeOD-*d*_4_), δ (ppm): 7.34 (t, 1H, *J* = 8 Hz, *Ph-CH*), 7.19 (d, 1H, *J* = 8 Hz, *Ph-CH*), 7.09 (s, 1H, *Ph-CH*), 6.98 (d, 1H, *J* = 8 Hz, *Ph-CH*), 3.80 (s, 2H, Ph*CH2*NH2); 3.50, 3.39 (2 × q, *J* = 6 Hz, N*CH*_2_CH_3_ rotamers); 3.09, 2.97 (2 × s, 3H, N*CH*_3_ rotamers); 1.26, 1.18 (2 × t, 3H, *J* = 8 Hz, NCH_2_*CH*_3_ rotamers). MS-ESI (*m*/*z*): 209.16 (M + 1)^+^.

##### 3-(2-Aminoethyl) phenyl ethyl methylcarbamate (**3c**)

It was synthesized from 3-(cyanomethyl) phenyl ethylmethylcarbamate (**2c**). The crude oil compound was purified by chromatography column (eluent MeOH/CH_2_Cl_2_/NH_3_, 49/49/2), affording the title pure compound (**3c**) as a beige oil. Yield = 35.0%. ^1^H NMR (400 MHz, MeOD-*d*_4_), δ (ppm): 7.35 (t, 1H, *J* = 8 Hz, *Ph-CH*), 7.13 (d, 1H, *J* = 8 Hz, *Ph-CH*), 7.02 (s, 1H, *Ph-CH*), 6.99 (d, 1H, *J* = 8 Hz, *Ph-CH*); 3.50, 3.39 (2 × q, *J* = 8 Hz, N*CH*_2_CH_3_ rotamers); 3.09, 2.97 (2 × s, 3H, N*CH*_3_ rotamers); 3.14 (t, *J* = 8 Hz, 2H, H_2_N*CH*_2_CH_2_Ph), 2.94 (t, *J* = 8 Hz, 2H, H_2_NCH_2_*CH*_2_Ph); 1.25, 1.17 (2 × t, 3H, *J* = 8 Hz, NCH_2_*CH*_3_ rotamers). MS-ESI (*m*/*z*): 223.18 (M + 1)^+^.

#### 2.2.3. General Procedure for the Synthesis of the Rivastigmine–Indole Hybrids (**5a1-3**; **5b1-3**; **5c1-3**)

To a water/ice-cooled solution of 1 mmol of an indole-carboxylic derivative (1*H*-indole-2-carboxylic acid (**4_1_**), 5-hydroxy-1*H*-indole-2-carboxylic acid (**4_2_**), 7-hydroxy-1*H*-indole-2-carboxylic acid (**4_3_**)) in dry DMF (5 mL), under nitrogen atmosphere, was added 2 mmol of NMM, followed by the addition of 1 mmol of TBTU, and this solution mixture was stirred for 50 min. Then, this activated indole-carboxylic acid solution was added dropwise to a water/ice-cooled solution of 1 mmol of the corresponding amine-carbamate derivative (3-aminophenyl-ethylmethylcarbamate (**3a**), 3-(aminomethyl)phenyl-ethylmethylcarbamate (**3b**) or 3-(2-aminoethyl)phenyl ethylmethylcarbamate (**3c**)) in 5 mL of dry DMF. The reaction mixture was stirred for 20 h under a nitrogen atmosphere, with the completion of the reaction controlled by TLC (AcOEt/hexane, 5/1). Afterward, DMF was evaporated under high vacuum. The residue was diluted with CH_2_Cl_2_, and the organic solution was washed with H_2_O, dried over anhydrous Na_2_SO_4_, filtered, roto-evaporated and dried under vacuum to obtain the crude solid, which was subsequently purified by chromatography column (eluent AcOEt/hexane, 5/1).

##### 3-[(1*H*-indole-2-carboxamido)phenyl ethyl(methyl)carbamate (**5a1**)

The title compound **5a1** was obtained as a white powder, according to the general procedure, using 3-aminophenylethylmethylcarbamate (**3a**) and 1*H*-indole-2-carboxylic acid (**4_1_**). Yield = 39.3%, M.P. = 214–216 °C. ^1^H NMR (400 MHz, MeOD-*d*_4_), δ (ppm): 7.65 (s, 1H, *Ar-CH*), 7.65 (d, 1H, *J* = 9 Hz, *Ar-CH*), 7.54 (d, 1H, *J* = 9 Hz, *Ar-CH*), 7.44 (d, 1H, *J* = 9 Hz, *Ar-CH*), 7.35 (t, 1H, *J* = 9 Hz, *Ar-CH)* 7.30 (s, 1H, *J* = 9 Hz, *Ar-CH*), 7.23 (t, 1H, *J* = 9 Hz, *Ar-CH*), 7.07 (t, 1H, *J* = 9 Hz, *Ph*), 6.86 (d, 1H, *J* = 9 Hz, *Ar-CH*); 3.49, 3.40 (2 × q, 2H, *J* = 6 Hz, N*CH_2_-CH*_3_ rotamers); 3.11, 2.99 (2 × s, 3H, *J* = 6 Hz, *NCH*_3_ rotamers); 1.27, 1.17 (2 × t, 3H, *J* = 6 Hz, NCH_2_*CH*_3_ rotamers). ^13^C NMR (400 MHz, MeOD-*d*_4_), δ (ppm): 160.73, 155.12, 139.68, 137.29, 132.05, 128.98, 127.74, 123.96, 121.50, 119.83, 117.05, 113.93, 111.79, 103.77, 43.79, 33.05, 32.79, 11.91, 11.14. HRMS (ESI) were calculated for C_19_H_19_N_3_O_3_ [M + H] 337.1426, found 337.1432.

##### 3-(5-Hydroxy-1*H*-indole-2-carboxamido)phenyl ethyl(methyl)carbamate (**5a2**)

The title compound **5a2** was obtained as a white powder, according to the general procedure, using 3-aminophenylethylmethylcarbamate (**3a**) and 5-hydroxy-1*H*-indole-2-carboxylic acid (**4_2_**). Yield = 45.1%, M.P. = 181–182 °C. ^1^H NMR (400 MHz, MeOD-*d*_4_), δ (ppm):7.63 (s, 1H, *Ar-CH*), 7.53(d, 1H, *J* = 8 Hz, *Ar-CH*), 7.34 (t, 1H, *J* = 8 Hz, *Ar-CH*), 7.29 (d, 1H, *J* = 8 Hz, *Ar-CH)*, 7.14 (s, 1H, *Ar-CH*), 6.97 (s, 1H, *Ph*), 6.86 (d, 1H, *J* = 8 Hz, *Ar-CH*), 6.82 (d, 1H, *J* = 8 Hz, *Ar-CH*); 3.51, 3.39 (2 × q, *J* = 8 Hz, N*CH*_2_CH_3_ rotamers); 3.10, 2.98 (2 × s, 3H, N*CH*_3_ rotamers); 1.27, 1.18 (2 × t, 3H, *J* = 8 Hz, NCH_2_*CH*_3_ rotamers). ^13^C NMR (400 MHz, MeOD-*d*_4_), δ (ppm): 160.90, 154.51, 151.66, 150.96, 139.62, 132.38, 131.19, 128.95, 128.32, 117.02, 115.08, 113.80, 112.29, 104.47, 103.10, 43.78, 33.15, 32.89, 12.03, 11.23. HRMS (ESI) calculated for C_19_H_19_N_3_O_4_ [M + H] 353.1376, found 353.1372.

##### 3-(7-Hydroxy-1*H*-indole-2-carboxamido)phenyl ethyl(methyl)carbamate (**5a3**)

The title compound **5a3** was obtained as a light pink powder, according to the general procedure, using 3-aminophenylethylmethylcarbamate (**3a**) and 7-hydroxy-1*H*-indole-2-carboxylic acid (**4_3_**). Yield = 53.0%, M.P. = 175–177 °C. ^1^H NMR (400 MHz, MeOD-*d*_4_), δ (ppm):7.68 (s, 1H, *Ar-CH*), 7.58 (d, 1H, *J* = 8 Hz, *Ar-CH*), 7.38 (t, 1H, *J* = 8 Hz, *Ar-CH*), 7.29 (s, 1H, *Ar-CH*), 7.16 (d, 1H, *J* = 8 Hz, *Ar-CH*), 6.94 (d, 1H, *J* = 8 Hz, *Ar-CH*), 6.91 (t, 1H, *J* = 8 Hz, *Ar-CH*), 6.67 (d, 1H, *J* = 8 Hz, *Ar-CH*); 3.53, 3.43 (2 × q, *J* = 8 Hz, N*CH*_2_CH_3_ rotamers); 3.14, 3.01 (2 × s, 3H, N*CH*_3_ rotamers); 1.29, 1.20 (2 × t, 3H, *J* = 8 Hz, NCH_2_*CH*_3_ rotamers). ^13^C NMR (400 MHz, MeOD-*d*_4_), δ (ppm): 160.80, 154.91, 151.67, 143.68, 139.63, 130.36, 129.47, 128.99, 127.95, 120.77, 117.04, 113.91, 112.91, 107.61, 104.89, 43.79, 33.17, 32.90, 12.04, 11.24. HRMS (ESI) calculated for C_19_H_19_N_3_O_4_ [M + H] 353.1376, found 353.1381.

##### 3-((1*H*-indole-2-carboxamido)methyl)phenyl ethylmethylcarbamate (**5b1**)

The title compound **5b1** was obtained as a white powder, according to the general procedure, using 3-(aminomethyl)phenyl-ethylmethylcarbamate (**3b**) and 1*H*-indole-2-carboxylic acid (**4_1_**). Yield = 40.9%, M.P. = 193–195 °C. ^1^H NMR (400 MHz, MeOD-*d*_4_), δ (ppm): 7.58 (d, 1H, *J* = 9 Hz, *Ar-CH*), 7.42 (d, 1H, *J* = 9 Hz, *Ar-CH*), 7.35 (t, 1H, *J* = 9 Hz, *Ar-CH*), 7.23 (t, 1H, *J* = 9 Hz, *Ar-CH*), 7.20 (s, 1H, *Ph*)*,* 7.11 (d, 1H, *J* = 9 Hz, *Ar-CH*), 7.09 (s, 2H, *Ar-CH*), 7.04 (t, 1H, *J* = 9 Hz, *Ar-CH*), 7.00 (d, 1H, *J* = 9 Hz, *Ar-CH*), 4.59 (s, 2H, NH*CH*_2_Ph); 3.47, 3.36 (2 × q, 2H, *J* = 6 Hz, N*CH_2_-*CH_3_ rotamers); 3.07, 2.95 (2 × s, 3H, *J* = 6 Hz, N*CH*_3_ rotamers); 1.23, 1.15 (2 × t, 3H, *J* = 6 Hz, NCH_2_*CH*_3_ rotamers). ^13^C NMR (400 MHz, MeOD-*d*_4_), δ (ppm): 162.79, 155.04, 151.60, 150.82, 140.65, 132.09, 131.04, 129.10, 128.29, 124.25, 120.61, 120.26, 114.70, 112.27, 104.44, 102.45, 43.78, 42.27, 33.16, 32.89, 12.01, 11.23. HRMS (ESI) calculated for C_20_H_21_N_3_O_3_ [M + H] 351.1583, found 351.1590.

##### 3-((5-Hydroxy-1*H*-indole-2-carboxamido)methyl)phenyl-ethyl(methyl)carbamate (**5b2**)

The title compound **5b2** was obtained as a white/beige powder, according to the general procedure, using 3-(aminomethyl)phenyl-ethylmethylcarbamate (**3b**) and 5-hydroxy-1*H*-indole-2-carboxylic acid (**4_2_**). Yield = 38.5%, M.P. = 190–191 °C. ^1^H NMR (400 MHz, MeOD-*d*_4_), δ (ppm): 7.34(t, 1H, *J* = 8 Hz, *Ar-CH*), 7.26 (d, 1H, *J* = 8 Hz, *Ar-CH*), 7.23 (d, 1H, *J* = 8 Hz, *Ar-CH*), 7.10 (s, 1H, *Ar-CH*), 6.99 (d, 1H, *J* = 8 Hz, *Ar-CH*), 6.93 (s, 2H, *Ar-CH*), 6.79 (d, 1H, *J* = 8 Hz, *Ar-CH*); 3.48, 3,37 (2 × q, 1H, *J* = 8 Hz, N*CH_2_CH*_3_ rotamers); 3.07, 2.95 (2 × s, 3H, *NCH*_3_ rotamers); 1.23, 1.15 (2 × t, 3H, *J* = 7 Hz, NCH_2_*CH*_3_ rotamers). ^13^C NMR (400 MHz, MeOD-*d*_4_), δ (ppm): 162.73, 154.98, 151.78, 140.78, 136.96, 130.68, 129.08, 127.64, 124.24, 123.69, 121.37, 120. 62, 119.77, 111.74, 103.14, 43.75, 42.28, 32.98, 32.86, 11.99, 11.20. HRMS (ESI) calculated for C_20_H_21_N_3_O_4_ [M + H] 367.1531, found 367.1535.

##### 3-((7-Hydroxy-1*H*-indole-2-carboxamido)methyl)phenyl-ethyl(methyl)carbamate (**5b3**)

The title compound **5b3** was obtained as a beige powder, according to the general procedure, using 3-(aminomethyl)phenyl-ethylmethylcarbamate (**3b**) and 7-hydroxy-1*H*-indole-2-carboxylic acid (**4_3_**). Yield = 51.2%, M.P. = 220–221 °C. ^1^H NMR (400 MHz, MeOD-*d*_4_), δ (ppm): 7.38 (t, 1H, *J* = 8 Hz, *Ph*), (7.28 (d, 1H, *J* = 8 Hz, *Ph*), 7.14, 7.11 (2 × bs, 3H, *Ph*), 7.02 (d, 1H, *J* = 8 Hz, *Ph*), 6.90 (t, 1H, *J* = 8Hz, *Ph*), 6.64 (d, 1H, *J* = 8 Hz, *Ph*), 4.62 (s, 2H, NH*CH*_2_Ph); 3.51, 3.39 (2 × q, 2H, *J* = 7 Hz, N*CH*_2_CH_3_ rotamers); 3.10, 2.98 (2 × s, 3H, *J* = 6 Hz, N*CH*_3_ rotamers); 2.24, 2.18 (2 × t, 3H, *J* = 6 Hz, NCH_2_*CH*_3_ rotamers). ^13^C NMR (400 MHz, MeOD-*d*_4_), δ (ppm): 162.70, 155.06, 151.44, 143.69, 140.75, 130.31, 129.82, 129.48, 127.55, 124.27, 120.64, 112.74, 107.36, 104.24, 43.69, 42.38, 32.86, 12.08, 11.24 HRMS (ESI) calculated for C_20_H_21_N_3_O_4_ [M + H] 367.1531, found 367.1525.

##### 3-(2-(1*H*-indole-2-carboxamido)ethyl)phenyl ethylmethylcarbamate (**5c1**)

The title compound **5c1** was obtained as a white powder, according to the general procedure, using 3-(2-aminoethyl)phenyl ethylmethylcarbamate (**3c**) and 1*H*-indole-2-carboxylic acid (**4_1_**). Yield = 34.1%, M.P. = 144–146 °C. ^1^H NMR (400 MHz, MeOD-*d*_4_), δ (ppm): 7.30–7.01 (m, 4H, *Ar-CH*), 6.90 (bs, 3H, *Ar-CH*), 6.76 (d, 1H, *J* = 8 Hz, *Ar-CH*), 6.49 (s, 1H, *Ar-CH*), 3.75 (bs, 2H, C*H*_2_NCO); 3.45, 3.41 (2 × q, 2H, *J* = 7 Hz, N*CH*_2_-CH_3_ rotamers); 2.98, 2.82 ((1s + 1bs), 5H, *N-CH*_3_ rotamers + NH*CH*_2_Ph); 1.22, 1.17 (2 × t, 3H, *J* = 7 Hz, NCH_2_*CH*_3_ rotamers). ^13^C NMR (400 MHz, MeOD-*d*_4_), δ (ppm): 165.34, 155.42, 151.59, 150.76, 131.29, 130.38, 128.97, 127.81, 125.74, 122.07, 119.62, 114.04, 112.11, 104.19, 102.76, 43.77, 33.18, 32.88, 12.03, 11.29. HRMS (ESI) calculated for C_21_H_23_N_3_O_3_ [M + H] 365.1739, found 365.1744.

##### 3-(2-(5-Hydroxy-1*H*-indole-2-carboxamido)ethyl)phenyl ethyl(methyl)carbamate (**5c2**)

The title compound **5c2** was obtained as a white/beige powder, according to the general procedure, using 3-(2-aminoethyl)phenyl ethylmethylcarbamate (**3c**) and 5-hydroxy-1*H*-indole-2-carboxylic acid (**4_2_**). Yield = 57.9%, M.P. = 114–115 °C. ^1^H NMR (400 MHz, MeOD-*d*_4_), δ (ppm):7.59 (d, 1H, *J* = 8 Hz, *Ar-CH*), 7.39 (d, 1H, *J* = 8 Hz, *Ar-CH*), 7.20 (t, 1H, *J* = 8 Hz, *Ar-CH*) 7.07 (d, 1H, *J* = 8 Hz, *Ar-CH*), 7.05 (d, 1H, *J* = 8 Hz, *Ar-CH*), 6.96 (bs, 2H, *Ar-CH*), 6.67 (s, 1H, *Ar-CH*), 3.81 (t, *J* = 8 Hz, *CH*_2_NCO); 3.49, 3.43 (2 × q, 2H, *J* = 8 Hz, N*CH*_2_CH_3_ rotamers); 3.09, 3.01 ((1s + 1bs), 5H, (N*CH*_3_ rotamers + NH*CH*_2_Ph)); 1.26, 1.21 (2 × t, 3H, *J* = 7 Hz, NCH_2_*CH*_3_ rotamers). ^13^C NMR (400 MHz, MeOD-*d*_4_), δ (ppm): 165.24, 155.21, 151.53, 136.17, 129.86, 129.08, 127.13, 125.75, 123.22, 122.19, 121.03, 119.57, 111.40, 103.39, 43.75, 33.17, 32.87, 12.02, 11.28. HRMS (ESI) calculated for C_21_H_23_N_3_O_4_ [M + H] 381.1689, found 381.1683.

##### 3-(2-(7-Hydroxy-1*H*-indole-2-carboxamido)ethyl)phenyl ethyl(methyl)carbamate (**5c3**)

The title compound **5c3** was obtained as a light beige/pink powder, according to the general procedure, using 3-(2-aminoethyl)phenyl ethylmethylcarbamate (**3c**) and 7-hydroxy-1*H*-indole-2-carboxylic acid (**4_3_**). Yield = 49.4%, M.P. = 161–163 °C. ^1^H NMR (400 MHz, MeOD-*d*_4_), δ (ppm):7.29 (bs, 1H, *Ar-CH*), 7.16 (bs, 1H, *Ar-CH*), 7.07 (d, 1H, *J* = 8 Hz, *Ar-CH*), 6.93 (bs, 2H, *Ar-CH*); 6.86 90 (t, 1H, *J* = 8 Hz, *Ar-CH*), 6.69 (s, 1H, *Ar-CH*), 6.59 (d, 1H, *J* = 8 Hz, *Ar-CH*), 3.77 (bs, 2H, *CH*_2_NCO); 3.47, 3.39 (2 × q, 1H, *J* = 8 Hz, N*CH*_2_CH_3_ rotamers); 3.06, 2.97 ((1s + 1bs), 5H, (N*CH*_3_ rotamers+ N*CH*_2_Ph)); 1.22, 1.17 (2 × t, 3H, *J* = 7 Hz, NCH_2_*CH*_3_ rotamers). ^13^C NMR (400 MHz, MeOD-*d*_4_), δ (ppm): 165.08, 155.15, 151.62, 143.41, 129.40, 129.01, 126.53, 125.80, 122.28, 120.44, 119.63, 112.42, 107.10, 104.12, 43.76, 33.19, 32.87, 12.06, 11.20. HRMS (ESI) calculated for C_21_H_23_N_3_O_4_ [M + H] 381.1689, found 381.1696.

### 2.3. Molecular Modeling

The new RIV-IND hybrids were docked in human acetylcholinesterase (hAChE) and human butyrylcholinesterase (hBChE) using two X-ray crystallographic structures downloaded from RCSB Protein Data Bank—PDB entries 4EY7 (complex of hAChE with donepezil) [27] and 4TPK (complex of hBChE with *N*-((1-(2,3-dihydro-1*H*-inden-2-yl)piperidin-3-yl)methyl)-*N*-(2-methoxyethyl)-2-naphthamide) [28]. These X-ray structures were selected due to the resemblance found between the original ligands in the crystallographic structures and the herein-studied RIV-IND hybrids. For the molecular docking studies, the enzyme models were prepared using only the single protein chain, removing the cocrystallized inhibitors as well as the crystallographic waters contained in the PDB structure, using Maestro v. 9.3 [29], and then hydrogen atoms were added. The RIV-IND models were created in Maestro, and minimum energy optimized structures were obtained by random conformational search using Ghemical v. 3.0.0 [30]. Lastly, the optimized structures of the compounds were docked into the cavity of the hAChE and hBChE model structures using Gold software v. 3.2. [31]. Goldscore was considered the best fitness function and 100 genetic algorithm steps were performed to obtain the finest correspondence between the RIV-IND hybrids and the original ligands of hAChE and hBChE (radius = 5). Each conformation was ranked according to its scores with the Goldscore scoring function. For each compound, the top 10 solutions were visually checked and critically evaluated. Figures illustrating the possible interactions between the RIV-IND ligands and the enzymes were obtained with UCSF Chimera software v. 1.6.2 [32], selecting, as an interest zone, the residues between 0 and 5.0 Å from the original position of the ligand in the crystal structure. The docking protocol used herein was validated by redocking the cocrystallized ligands (original ligands) into the active sites of the corresponding enzymes [27,28], leading to their best match (superimposition) with the crystallographic ligands with a root-mean-square deviation (RMSD) until 1 Å (1.093 for AChE and 0.765 for BChE) (see Appendix A in Supplementary Material).

### 2.4. Radical Scavenging Activity

The radical scavenging activity (antioxidant capacity) of the herein-developed RIV-IND hybrids was determined by the DPPH method [33]. Working solutions of the compounds in 10% DMSO/MeOH (, HoneywellSeelze, Germany) medium were used to prepare 5 methanolic solutions (5 points, 3 replicates) containing DPPH (Sigma-Aldrich, Steinheim, Germany) and different concentrations of compound up to 3.5 mL total volume. Each calibration point was kept for 30 min under darkness, followed by measurement of the absorbance value at 517 nm. The respective antioxidant activity (%AA) was calculated using the following equation:$$\%AA=\frac{A_{DPPH}-A_{ligand}}{A_{DPPH}}\times 100$$

From the plot of %AA versus ligand concentration, the EC_50_ (concentration of the ligand corresponding to %AA = 50) was determined. The reported results were obtained as the mean values of three independent experiments.

### 2.5. Cholinesterase Inhibition

The enzyme inhibition of electric eel acetylcholinesterase (eeAChE) and equine butyrylcholinesterase (eqBChE) by our compounds was evaluated through an adaptation of Elmman´s method [24,33] using reagents from Sigma-Aldrich (Steinheim, Germany). Stock solutions of each compound (1 mg) in MeOH (1 mL) were prepared and then working solutions for each ligand were obtained by the corresponding dilution of the stock solutions. A control measurement with 5 replicates (without ligand) and a calibration curve of 5 points with 3 replicates per point was performed. The preparation of each solution involved fixed volumes of 4-(2-hydroxyethyl)-1-piperazine-ethanesulfonic acid (HEPES) and cholinesterase (eeAChE or eqBChE) with different volumes of ligand solution, and it was left to rest for 15 min immediately after adding the cholinesterase. A blank solution containing HEPES and MeOH was also prepared at the same time. After 15 min of reaction, acetylcholine iodide (AChI) or butyrylcholine iodide (BChI) and 5,5′-dithiobis-(2-nitrobenzoic acid) (DTNB) were added to both solutions in spectrophotometer cuvettes, and the absorption signal at 405 nm, with a slit width of 1 nm, was recorded during 5 min. The slope for each calibration point (absorbance versus reaction time) was determined and the inhibition for each compound was calculated by using the following equation:$$\%Inhibition=100-(\frac{v_{i}}{v_{0}}\times 100)$$
where *v*_i_ is the reaction rate (slope) for each point of the calibrate with ligand and *v*_0_ is the initial reaction rate (slope) for the control point (without ligand). When the inhibition values of the 5 points of the calibrate and their 3 replicates were obtained, the % inhibition was plotted versus the concentration of ligand, and the IC_50_ value (inhibitor concentration corresponding to 50% inhibition of the enzyme) was calculated. Experiments were performed twice for each compound.

### 2.6. Inhibition of Self Aβ_1-42_ Aggregation

The Aβ peptide (Genecust, Boynes, France) was treated with 1,1,1,3,3,3-hexafluoropropan-2-ol (HFIP, Aldrich, Steinheim, Germany) and dissolved in a CH_3_CN/Na_2_CO_3_ (300 µM)/NaOH (250 µM) (48.3:48.3:4.3, *v*/*v*/*v*) mixture in order to have a stable stock solution of Aβ_1-42_ with concentration 500 µM.

Solutions of the assayed compounds (0.5 mg/mL) were firstly prepared in 60% (*v*/*v*) DMSO/MeOH medium and then diluted in phosphate buffer (0.215 M, pH 8.0) to a concentration of 120 µM. The Aβ_1-42_ aggregation inhibition assays were performed according to a reported method based on the fluorescence emission of thioflavin T (ThT, Aldrich Steinheim, Germany) [15,34,35]. Therefore, Aβ_1-42_ (40 µM) was incubated at 37 °C for 24 h in phosphate buffer and in the presence or absence of each ligand (20 µM). Then, the samples were added to a 96-well plate with 180 µL of 5 µM ThT in 50 mM glycine-NaOH (pH 8.5) buffer. Blank samples were identically prepared for each concentration but in the absence of the peptide. The ThT fluorescence was measured at 445 nm (excitation) and 485 nm (emission). The percent of aggregation inhibition was calculated by the following equation:I% = 100 − (IF_i_/IF_0_ × 100),
in which IF_i_ and IF_0_ are the fluorescence intensities, in the presence and the absence of the tested compound, subtracted from the fluorescence intensities due to the respective blanks. The reported values were obtained as the mean ± SEM of two different experiments performed in duplicate.

### 2.7. Cell Viability and In Vitro Neuroprotection

Cell viability, following treatment with the compounds, was determined using the colorimetric MTT (3-(4,5-dimethylthiazol-2-yl)-2,5-diphenyltetrazolium bromide) assay [36]. The tested compounds (**5a1**-**3**, **5b1-3**, **5c1-3**) were dissolved in DMSO to produce stock solutions of 25 mM and stored at −20 °C. A concentration screening was performed for 0.5 µM to 20 µM final concentrations to choose the highest nontoxic concentration. SH-SY5Y human neuroblastoma cell line (ATCC-CRL-2266) was cultured in Dulbecco’s modified Eagle’s medium (DMEM) (Gibco-Invitrogen, Life Technologies Ltd., Waltham, MA, USA) with heat-inactivated fetal calf serum (10%), penicillin (50 U/mL) and streptomycin (50 µg/mL) at 37 °C with 5% CO_2_. Cells were plated at 0.15 × 10^6^ cells/mL one day prior to the start of the experiment. The cells were then incubated with the appropriate concentration of the compounds for 24 h. After the incubation period, the cells were washed once with PBS and incubated with 200 µL MTT (0.5 mg/mL) for 2 h at 37 °C with 5% CO_2_. During this period, cellular dehydrogenase successfully metabolized MTT into formazan [36], which was then solubilized using 200 µL of 0.04 M HCl/isopropanol. Absorbance was measured at 570 nm.

To test whether the compounds have a protective effect against toxicity induced by Aβ_1-42_ or iron/ascorbate, the cells were pre-incubated for 1 h with each compound, followed by the addition of Aβ_1-42_ or iron/ascorbate and incubation for an additional 24 h. Aβ_1-42_ (Bachem, Torrance, CA, USA) was prepared in sterile water to produce a stock solution of 443 µM and added to the cells at a final concentration of 1 µM. Ferrous sulfate and L-ascorbic acid (Sigma Chemical Co, St. Louis, MO, USA), prepared in sterile water, were added to the medium together at a final concentration of 5 mM and 2.5 mM, respectively. For this assay, the optimal concentration of each compound was used to prevent cellular demise induced by Aβ_1-42_ or iron/ascorbate. Compound **5a1** was added to the medium at a 1 µM final concentration; compounds **5a2**, **5a3**, **5b1**, **5b2** and **5c2** were added to the medium at a 2.5 µM final concentration; compounds **5c1** and **5c3** were added to the medium at a 5 µM final concentration; and compound **5b3** was added to the medium at a 10 µM final concentration. The DMSO final concentration did not exceed 0.05% (*v*/*v*), and cells were monitored for possible alterations, which did not occur. Controls were employed for each plate using untreated cells and Aβ_1-42_ or iron/ascorbate only. Cell reduction ability was normalized to the untreated condition. All data are expressed as mean ± SEM of at least three independent experiments performed in duplicate. Statistical analyses were performed for normality assessment using Shapiro–Wilk’s test followed by the Kruskal–Wallis test considering that normality was not verified. Post hoc analysis was performed using uncorrected Dunn’s test. A *p* value < 0.05 was considered statistically significant.

### 2.8. Prediction of Pharmacokinetic Properties

The drug-likeness properties of the developed hybrid compounds were predicted in silico. Previously, the model compounds were built in Maestro v. 9.3 [29] and their structure was optimized by energy minimization (see Section 2.3). Then, the program QikProp v. 2.5 [37] was used to determine several pharmacokinetic properties and descriptors, such as polar surface area (PSA), octanol/water partition coefficient (clog *P*_o/w_), interaction with human albumin, brain–blood barrier (BBB) permeability, CNS activity, Caco-2 and MDCK cell permeability, human oral absorption and violations of Lipinski’s rule.

## 3. Results and Discussion

### 3.1. Synthesis of the RIV-IND Hybrids

The overall synthesis route to obtain the new target nine compounds (**5a1-3**, **5b1-3**, **5c1-3**) is displayed in Figure 2. Firstly, three amino (alkyl) rivastigmine derivatives (**3a-c**) were prepared, following a two-step reaction procedure. The first step involved the phenol carbamylation of the nitro- and cyano(alkyl)-phenols (**1a-c**) through their reaction with *N*-ethyl-*N*-methylcarbamoyl chloride in dry triethylamine (TEA), under reflux, to obtain the expected carbamoyl derivatives (**2a-c**) with almost quantitative yields. The second step involved the hydrogenolysis of the nitro and nitrile group of these intermediates, which was carried out in methanol and a H_2_ atmosphere (4 atm), to give the corresponding amino derivatives (**3a-c**). Finally, these free amino-rivastigmine intermediates (**3a-c**) were condensed with three indoyl carboxylic acid derivatives (**4_1_**, **4_2_**, **4_3_**) via a peptide bond formation, using a coupling reagent, 2-(1*H*-benzotriazole-1-yl)-1,1,3,3-tetramethylaminium tetrafluoroborate (TBTU), in anhydrous DMF under a N_2_ atmosphere, affording the corresponding final nine RIV-IND hybrid compounds (**5a1-3**, **5b1-3**, **5c1-3**).

### 3.2. Molecular Modeling Studies

Among several pathophysiological targets associated with AD, the cholinergic pathway has been the most adopted for the FDA-approved anti-AD drugs to compensate for the decrease in the acetylcholine (ACh) neurotransmitter, with a corresponding decline in memory and learning capacity. This cholinergic neuron degeneration is due to serine hydrolases AChE and BChE, which seem also to be implicated in Aβ neurotoxicity and plaque maturation. In fact, while AChE is selective for ACh hydrolysis, BChE can also accommodate and degrade many other substrates such as several neuroactive peptides [38].

Both human enzymes (hAChE, hBChE) have ca 65% amino acid sequence homology, a catalytic active site (CAS) positioned at the bottom of the hydrophobic gorge (20 Ẳ deep) and a peripheral anionic site (PAS) at the rim of the gorge [39,40]. Although closely related, these enzymes have quite different substrate specificities, mainly due to the different number/type of aromatic residues lining the active site gorge, which results in different gorge shapes/sizes. In particular, the wider shape of the hBChE gorge allows better accommodation of bulkier compounds and their accessibility to the active site [41,42]. The active sites of both enzymes are composed of a catalytic triad (serine, histidine and glutamate residues), an acyl binding pocket and a choline binding site.

In this context, envisaging some anticipation or rationalization of the experimental results obtained for the inhibition of both cholinesterases by our compounds, molecular modeling docking studies with Gold software v. 5.2. (CCDC, Cambridge, UK) [31] were performed for all the RIV-IND hybrids, and a summary of the most relevant results is presented herein.

The docking poses found for the selected compounds **5a3** and **5c2**, inside the active site cavity of hAChE, are shown in Figure 1. This figure shows that the RIV moiety of **5a3** and **5c2** is inside the CAS, while the IND portion is on the PAS side. Moreover, compound **5c2** has a twisted position of the BIM moiety relative to all the RIV-IND hybrids, forming two H bonds, one between the hydroxyl group of IND and Ser293 (1.9 Å) and another between the *O*-carbonyl atom adjacent to IND and Tyr124 (3.2 Å).

All the compounds present similar accommodation within the active site when compared with the original ligand, but compound **5c2** has the IND moiety twisted. The hybrids are positioned close to some relevant residues of hAChE such as Tyr124, Trp286, Tyr72, Tyr341, Trp86, Tyr337, Phe338, Glu202 and Ser293. They all have the RIV moiety inside the CAS and the IND moiety orientated towards the PAS. While compounds **5a1**-**3**, **5b1**, **5b3** and **5c3** have no detected H-binding interactions, compounds **5b2** and **5c1**-**2** present one (**5b2**, **5c1**) and two H bonds (**5c2**, see Figure 1b)). These H-bond interactions are established between the hydroxyl group of IND and Ser293 (2.3 Å) for **5b2** and between the NH group adjacent to the IND portion and Asp74 (1.3 Å) for **5c1**. Compounds **5a3**, **5c1** and **5c3** are also able to establish π-π stacking interactions between the IND group and Trp286, as well as between the phenyl group of RIV and Phe338 (**5a3**, **5c3**) or Tyr337 (**5c1**). Donepezil (DNP), the original ligand in the crystal structure of the complex with recombinant hAChE (PDB code 4EY7), is a powerful AChE inhibitor (IC_50_ = 7.5 nM [24]), presenting a π-π stacking interaction of the aromatic groups (indanone in PAS and benzyl in CAS) and also a H bond between the indanone oxygen atom and Phe295 [27]. Among the herein-studied RIV-IND compounds, both **5a3** and **5c3** evidence the best superimposition with the original ligand contained in the AChE complex.

Concerning the docking results for the hBChE, Figure 2 presents the obtained poses for two representative compounds (**5a2** and **5b1**). All the RIV-IND compounds are accommodated in the enzyme active site, showing that the IND moiety is in the CAS and the RIV moiety is nearer the PAS, although not installed so highly in the rim of the gorge as the original ligand (see Figure 2 and Appendix A)). Nevertheless, the compounds are potentially able to establish binding interactions with reference active residues of the enzyme such as Trp82, Phe329 and Trp231 28].

In the original ligand (from the crystal structure corresponding to PDB code 4TPK), the naphthalene moiety is situated in the acyl binding pocket, establishing a π-π stacking interaction with Trp231, while the 1H-indene moiety is positioned over residues Ile69 and Asp70 in the PAS portion. Concerning the herein-studied compounds, **5a2** is able to establish two H bonds within the CAS, namely between the linker NH amidic group and Ser198 (2.1 Å) and between the hydroxyl substituent of the IND moiety and Leu286 (1.4 Å), while **5b1** forms only one H bond between ether oxygen of the RIV moiety and Thr120 (2.7 Å). Finally, the remaining compounds interact with the residues of BChE by establishing a variable number of H-bonds within the active center, namely one for **5a1** (between the NH amidic group of the linker and Ser198 (1.9 Å)); one for **5c1** (between ether oxygen of the RIV moiety and Thr120 (2.6 Å)); two for **5a3** (between the NH group of the linker and Ser198 (2.1 Å) and between the hydroxyl of IND and Ser287 (2.0 Å)), **5b2** (between the carbonyl oxygen of the IND moiety and Ser198 (3.4 Å) and the NH group of IND and Ser198 (1.6 Å)), **5c2** (between the NH group of IND and Ser198 (1.7 Å) and between the ether oxygen and Thr120 (2.9 Å)) and **5c3** (between the NH group of IND and Ser198 (1.6 Å) and between the ether oxygen and Thr120 (3.0 Å); and finally, three for **5b3** with Ser198 (and NH of IND (1.7 Å) or the hydroxyl group of IND (2.6 Å) or the carbonyl oxygen of IND (3.2 Å)). Moreover, compounds **5b2**-**3** and **5c1**-**3** seem also to be able to establish π-π stacking interactions between the phenyl group of the RIV moiety and Trp82.

Appendix A contains the Goldscore scoring function values (63.37–72.82) obtained for the RIV-IND compound; these values increase with the rise in the linker length.

Interestingly, a comparison of the herein obtained results with those previously found for other derivatives of rivastigmine (RIV-BIM) [24] shows a similar spatial orientation of the compounds inside the active site of AChE, since all the RIV-BIM hybrids also had the RIV moiety inside the CAS, while the BIM portion is orientated to the entrance of the gorge. Moreover, all the herein-studied RIV-IND hybrids present the RIV portion directed towards the PAS of BChE, such as already happened with the twisted compounds of the RIV-BIM series.

### 3.3. Free Radical Scavenging Activity

The new RIV-IND compounds were also screened for their radical scavenging activity (EC_50_), according to the 2,2-diphenyl-1-picrylhydrazyl (DPPH) free radical method. This assay relies on the compounds being able to scavenge DPPH, therefore decolorizing (yellow color) the DPPH methanolic solution (violet color) by its reduction [43]. The results are presented in Table 1, which shows that all the hybrids containing a hydroxyl indole substituent (**5a2**-**3**, **5b2**-**3**, **5c2**-**3**) show a good radical scavenging activity, with EC_50_ values ranging from 7.8 to 20.7 µM, much higher than that of the hydroxypiridinone drug deferiprone (148 µM) and close to (**5a3**, **5b3**, **5c3**) or even better (**5a2**, **5b2**, **5c2**) than strong antioxidants such as Trolox (13.8 µM) or vitamin C (15 µM) [44]. In fact, the hydroxyl group is expected to have an important role in the inhibition of the radical formation as well as in its capture [45]. Moreover, the capacity of these hydrogen-providing groups seems also to be dependent on the position of the hydroxyl group in the IND moiety of the hybrids, with compounds **5a2**, **5b2** and **5c2** revealing to be more active than their analogs **5a3**, **5b3** and **5c3**, respectively.

As expected, the compounds without a labile hydrogen atom of the indole hydroxyl group, such as **5a1**, **5b1** and **5c1**, show much lower activity (EC_50_ > 2 mM) than those containing that group.

### 3.4. Inhibition of Cholinesterases

The inhibitory capacity of the developed RIV-IND hybrids and the reference anti-ChE compound rivastigmine (RIV) was evaluated against the electrophorus electricus acetylcholinesterase (eeAChE) and the equine butyrylcholinesterase (eqBChE) through an adaptation of the Elmman´s method [24,33]. The obtained results, expressed as IC_50_ values, are depicted in Table 1, which indicates that the best AChE inhibitor is **5a3** (IC_50_ = 10.9 µM), followed by **5c3** (IC_50_ = 26.8 µM), both showing improvement relative to RIV (IC_50_ = 32.1 µM), while the best two BChE inhibitors are **5a2** (IC_50_ = 3.22 µM) and **5b3** (IC_50_ = 5.7 µM), which present lower activity than RIV (IC_50_ = 0.39 µM).

The results obtained for AChE inhibition show a dependence on the size of the linker for the hybrids **5a1**-**2**, **5b1**-**2** and **5c1**-**2**, namely increasing with its length. Interestingly, the docking studies suggested that compounds **5a3** and **5c3** have the best superimposition with the original ligand; also, compounds **5a1**-**2** and **5b1** do not exhibit H-bond interactions, while compound **5c2** has the IND moiety twisted relative to all the other hybrids. These results evidence that in addition to the length of the compound and its spatial position inside the active site of the enzyme, the existence of H bonds can also be a determining factor. The same linker size-dependent behavior is not observed for compounds **5a3**, **5b3** and **5c3**, for which the AChE inhibitory capacity follows the order **5a3** > **5c3** > **5b3**, probably due to the fact that **5a3** and **5c3** have the best superimposition with the original ligand and are also able to establish π-π stacking interactions involving both the IND group as well as the phenyl group of the RIV moiety.

Regarding BChE inhibition, it does not seem to be dependent on the linker size but on the number of established H-bonds (e.g., three for **5b3**, two for **5a2**-**3**, **5c2**-**3** and one for **5b1** and **5c1**) as well as the possible π-π stacking interactions (**5b2**-**3** and **5c1**-**3**) suggesting higher possible interactions with the enzyme. Therefore, these features positively influence the BChE inhibitory activity, which seems also to be associated with a unique spatial orientation for all the compounds, with the RIV moiety orientated in all the cases to the PAS and the IND portion inside the CAS. Moreover, since the active site of this enzyme is wider than that of AChE, there is also an increase in the variety of possible poses for the compounds, allowing the establishment of convenient interactions with the enzyme residues, and so the obtained results for BChE inhibition contained in Table 1 can also reflect this possibility.

Overall, concerning the inhibitory activity of the herein-studied RIV-IND hybrids, although they have somehow lower capacity than the RIV-BIM analogs [24], almost all of them show good results and present a moderate/low selectivity towards BChE, as the parent RIV drug (SI(RIV) = 82.3), the most selective ones being **5a2** (SI = 64.0) and **5b3** (SI = 14.2). So, the developed RIV-IND hybrids can keep the dual inhibitory potential of the parent rivastigmine, therefore acting in the early stages of AD (namely through AChE inhibition) as well as during the progression of the disease (BChE contribution is more important) [38].

### 3.5. Inhibition of Aβ_1-42_ Self-Aggregation

The inhibitory capacity of the RIV-IND hybrids towards Aβ_42_ self-aggregation was evaluated in vitro by molecular fluorescence spectroscopy through the thioflavin T (ThT) method [15,34,35]. In fact, since there is an increase in the absorbance and emission of ThT when it binds to amyloid fibrils, it is possible to determine the percent of Aβ_42_ self-aggregation (see Table 1). The assays were performed in the presence of 20 µM of inhibitor to overcome the solubility limitations of some compounds in the phosphate buffer aqueous medium.

Analysis of the data contained in Table 1 allows us to conclude that the presence and the position of the OH substituent groups in the IND moiety are important for the anti-amyloidogenic activity of the RIV-IND hybrids herein studied. In fact, the most effective compounds for the self-aggregation inhibition of Aβ_42_ are **5a3**, **5b3** and **5c3** (47.8–55.5%), which have the hydroxyl substituent group in the *ortho* position relative to the NH of the IND unit. These compounds present good inhibitory potential towards Aβ_42_ aggregation, with values analogous to those of the previously developed RIV-BIM compounds, although weaker than the selected reference curcumin (65.7%) [24]. The remaining compounds have no hydroxyl groups (**5a1**, **5b1**, **5c1**) or *para*-positioned hydroxyl groups (**5a2**, **5b2**) relative to the NH of IND and are more modest inhibitors of the aggregation (19.9–29.4%), with the exception of **5c2** (49.9%). Finally, preliminary studies on the inhibition of Cu-induced Aβ_42_ aggregation show higher inhibitory abilities for the chelators **5a3**, **5b3** and **5c3** (ca 64–84%), therefore pointing to some activity dependence on the capacity of these compounds for copper chelation.

### 3.6. Cell Viability and Neuroprotection

The new RIV-IND hybrids were tested to determine their protective role in the toxicity induced by either Aβ_1-42_ peptides or iron/ascorbate. SH-SY5Y cells were incubated with a set of concentrations of each of the compounds and a dose–response curve was constructed to select the highest nontoxic concentration. This dose was found to be 1 µM for **5a1** and **5b1** (when used with iron/ascorbate); 2.5 µM for **5a2**, **5a3**, **5b1** (when used with Aβ_1-42_), **5b2** and **5c2**; 5 µM for **5c1** and **5c3**; and 10 µM for **5b3** (Figure 3).

Despite the complexity of factors involved in AD pathogenesis, the accumulation of senile plaques composed of Aβ aggregates, including the more toxic Aβ_1-42_ form found in the core of the plaque, together with the production of reactive oxygen species (ROS) are key hallmarks of AD etiology [46,47]. Accordingly, we observed a significant decrease in cell viability following incubation with Aβ_1-42_ (Figure 4) and with iron/ascorbate (Figure 5) with an average of 69% and 67%, respectively. Interestingly, compounds **5b1**, **5b2** and **5c2** were able to rescue Aβ_1-42_-induced toxicity to levels that were not significantly different from untreated cells, although **5a3** and **5c3** also present neuroprotection (Figure 4). Regarding toxicity induced by ROS upon treatment with iron/ascorbate, only compound **5b3** was able to significantly increase cell viability after the trigger, though compounds **5a3** and **5b1** also present a neuroprotective effect (Figure 5).

### 3.7. Prediction of Pharmacokinetic Behavior

Some relevant pharmacokinetic properties of the RIV-IND hybrids and also the drug rivastigmine were predicted in silico (QikProp v. 2.5 [37]) and are summarized in Table 2. All the hybrids have good predicted values for molecular weight, PSA (van der Waals surface area of polar nitrogen and oxygen atoms), octanol/water partition coefficient, human albumin serum protein binding, brain–blood barrier permeability and MDCK permeability. Concerning Caco-2 cell permeability, all the compounds also have good values, though the hybrids without hydroxyl substituent groups (**5a1**, **5b1**, **5c1**) present excellent results (975-1104 vs. 1381 nm/s for rivastigmine). Furthermore, all the compounds but those with a hydroxyl group in the *para* position relative to the NH group of the IND moiety (**5a2**, **5b2**, **5c2** with 78–79%) have high oral absorption (>80%). All the compounds present 0 violations of Lipinski´s rule of five, therefore confirming their potential as oral drug candidates.

Comparing the predicted values contained in Table 2 with those of the rivastigmine parent anti-AD drug, the introduction of the IND moiety seems to affect mainly the Caco-2 and MDCK permeabilities by lowering them, also implying some decrease in the BBB permeability.

Although good pharmacokinetic descriptors and drug-likeness results are herein obtained, further studies must be conducted in terms of ADMET parameters as well as experimental assays to evaluate the effective BBB permeation and determine their metabolism and mechanism of elimination.

## 4. Conclusions

Due to the complexity of factors associated with the development and progression of AD, a multitarget approach for the development of potential new drugs has been adopted consistently in recent years. Herein, a series of nine RIV-IND hybrids was designed, synthesized and evaluated for several biological properties and free radical scavenging activity in view of their potential as multitarget anti-AD drugs. The molecular docking of these hybrids on the active center of the cholinesterases (AChE and BChE) demonstrated that the compounds are able to establish interactions with important residues of both enzyme active sites, thus giving support to their inhibitory capacity towards these enzymes, which may have a greater disease-modifying impact relative to the AChE-selective FDA-approved drugs. The in vitro determined cholinesterase inhibitory activity was found to be good for all the RIV-IND hybrids, with a moderate/low selectivity towards BChE, as the parent rivastigmine drug, and a dependence on the linker size for AChE inhibition. Concerning the inhibition of Aβ_42_ self-aggregation, it can be concluded that the presence and position of the OH groups in the IND moiety are important for the anti-amyloidogenic activity of the RIV-IND hybrids, the most effective ones being those that have the hydroxyl substituent group in *ortho* position relative to the NH of the IND portion. Regarding the free radical scavenging activity, compounds with hydroxyl substituents showed good results, particularly those *para-*positioned relative to the NH of the IND portion. Compounds **5b2** and **5c2** were able to prevent the toxicity induced by Aβ_1-42_ to neuroblastoma cells (SH-SY5Y), but only compound **5b3** was able to significantly protect cell viability after iron/ascorbate oxidative insult. The in silico evaluation of the drug-likeness of the hybrids revealed that all of them seem to have potential oral availability.

Overall, the activity of these compounds is not especially high, but compounds **5a3** and **5c3** combine the best results for a range of anti-AD hallmarks herein evaluated, appearing as leads in a preliminary characterization of compounds with possible anti-AD profiles.

## Data Availability

The data presented in this study are available in this article (and Appendix A).

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
