# Peer review of "New Multitarget Rivastigmine–Indole Hybrids as Potential Drug Candidates for Alzheimer’s Disease"

_pharmaceutics, 2024, doi:10.3390/pharmaceutics16020281_

Round 1

Reviewer 1 Report

Comments and Suggestions for Authors

Comments to the authors:

The manuscript entitled “New Multitarget Rivastigmine-Indole Hybrids as Potential Drug Canditates for Alzheimer´s Disease” is medicinal research supported by biological and in silico studies. The manuscript may be considered for publication; however, some minor revisions are required. More specifically, I expect the authors to address the following points:

1.      The manuscript has some typographical and spacing errors. Therefore, it is essential that the authors thoroughly check for them.

For example; degrees Celsius symbol must be corrected in all parts of the manuscript

2.      The manuscript lacks key references for all the software and modules used. Authors must cite or acknowledge the software’s utilized (main citation), and also check that all references are properly assigned.

3.      I recommend that authors double-check the values provided in Table 2 (Mol. Weight, PSA, clog, log K, and log BB) and present them in a proper way.

4.      Conclusions are too long and inadequate. I advise that authors confine their conclusions to key points.

Author Response

Answers to the Referees

Response to comments/suggestions of Referee 1 (in red color)

  1. The manuscript has some typographical and spacing errors. Therefore, it is essential that the authors thoroughly check for them. For example; degrees Celsius symbol must be corrected in all parts of the manuscript

The MS has been revised for typographical (including degrees Celsius symbol) and spacing errors.

  1. The manuscript lacks key references for all the software and modules used. Authors must cite or acknowledge the software’s utilized (main citation) and also check that all references are properly assigned.

      According with this query, the references for software were all properly verified, in format and assignment, and a new reference for UCSF Chimera software (ref. [32]) was added.

  1. I recommend that authors double-check the values provided in Table 2 (Mol. Weight, PSA, clog, log K, and log BB) and present them in a proper way.

As requested by the Reviewer, the values provided in Table 2 were double-checked and corrected to present them in a proper way (replacing comma by dot).

  1. Conclusions are too long and inadequate. I advise that authors confine their conclusions to key points.

Thank you, conclusions were revised according to suggestion.

Reviewer 2 Report

Comments and Suggestions for Authors

Some comments to the paper are reported below for improving the quality of the paper

-introduction Aducanumab, Lecanemab, Donanemab are not currently approved by EMA not only for the adverse effects but also for the questionable efficacy, please correct the row 51-52

-when you mentioned MTDL please cited 10.1016/j.ejmech.2015.10.001

-NMR spectra should be added as supllementary materials

-concerns about the docking prediction. Authors showed the superposition of original ligand crystallized vs docked pose. Based on the observation the redocking procedure produced poses in which the crystallized ligands were not correctly accommodated. redocking is a validation of the docking protocol and usually the RMSD between the crystallized and docked pose should be under 2 A. Authors should calculate the RMSD to assess the validity of the docking approach. After that. I suggest to rethink the docking output. In fact, I suggest to remove the original ligand from the docking output of the novel compounds while maintaining the superposition for the poses between the crystallized and docked original ligand. In the current way the pictures are confused and were not meaningful. After that there is no mention about the treatment of protein before docking and the details about grid and docking parameters. It is warmly recommended to perform MD simulation and deltaG calculation for the most promising compounds compared to the original ligand.

-the number of independent experiments for the in vitro tests should be reported.

-although the activity is not so significant, the paper has some interesting hints and the experimental part is in line with a preliminary characterization of compounds with possible antiAD profile.

Comments on the Quality of English Language

moderate changes are required

Author Response

Response to comments/suggestions of Referee 2 (in red color)

- Introduction Aducanumab, Lecanemab, Donanemab are not currently approved by EMA not only for the adverse effects but also for the questionable efficacy, please correct the row 51-52 .

Thank you for the information, that was introduced accordingly.

- When you mentioned MTDL please cite 10.1016/j.ejmech.2015.10.001.

Thank you for the suggestion of the reference citation, which is now ref.16.

- NMR spectra should be added as supplementary materials

The 1HNMR and 13CNMR spectra of the final products have been added as  supplementary information (Fig. S1 and Fig. S2, respectively).

- Concerns about the docking prediction. Authors showed the superposition of original ligand crystallized vs docked pose. Based on the observation, the redocking procedure produced poses in which the crystallized ligands were not correctly accommodated. Redocking is a validation of the docking protocol and usually the RMSD between the crystallized and docked pose should be under 2 A. Authors should calculate the RMSD to assess the validity of the docking approach. After that, I suggest rethinking the docking output. In fact, I suggest removing the original ligand from the docking output of the novel compounds while maintaining the superposition for the poses between the crystallized and docked original ligand. In the current way, the pictures are confused and were not meaningful. After that, there is no mention about the treatment of protein before docking and the details about grid and docking parameters. It is warmly recommended to perform MD simulation and deltaG calculation for the most promising compounds compared to the original ligand.

The authors are thankful to the Referee since with this query the docking studies were highly improved.

Redocking of the original ligands was performed again for both enzymes and the scoring function was changed from Asp to Goldscore. RMSD values until 1 Å (1.093 for AChE and 0.765 for BChE) were obtained (see experimental part, section 2.3). Figure S3 (in supplementary material) contains the superimposition of the co-crystallized ligands and their docked poses in the active sites of AChE and BChE.

In the experimental part, section 2.3, the performed treatment of the protein before docking is now mentioned as well as the procedure adopted to choose the ranked ligand poses.

Due to the change in scoring function, docking was performed again for all the compounds and so now, in section 3.2., new figures are presented with examples of docked compounds alone and the text is rewritten. Some changes in the text of section 3.4. were also done, in accordance with the new results obtained from MM.

The authors also thank the referee for the suggestion of enriching the herein presented results from MD with grid-based scoring and free energy calculations, though we are not able to produce these results with the GOLD software v. 3.2. Nevertheless, we are planning to adopt other docking programs in next publications, therefore allowing a more complete output of MD studies.

- The number of independent experiments for the in vitro tests should be reported.

The authors thank the Referee suggestion and now the number of independent experiments for all the in vitro tests are included in the present form of the MS, namely in the experimental part, footnote of Table 1 as well as in the captions of Figures 3, 4 and 5.  

- Although the activity is not so significant, the paper has some interesting hints and the experimental part is in line with a preliminary characterization of compounds with possible antiAD profile.

The authors agree with the referee and this point is now addressed in the Conclusions part.

Reviewer 3 Report

Comments and Suggestions for Authors

The multitargeting drug development of Alzheimer’s Disease has a broad interest to the readers. The authors designed and synthesized the nine rivastigmine-indole (RIV-IND) hybrids and investigated their biological properties with experiment and theory. As expected, the designed compounds might bind to the two serine hydrolases AchE and BChe and show inhibition of the target enzymes with the experiment. Also, the authors examined the antioxidant activity, inhibition of Ab1-42 aggregation, cell viability, and neuroprotection of the designed compounds. The author's discoveries in this work are quite interesting. Using molecular docking studies, the authors tried to interpret the experimental results. However, I think some points need to be addressed before the publication. The following lists are the questions and recommendations for improving the manuscript.

1.      The authors showed that designed compounds bind to the active site of two enzymes. With the docking results, they interpreted the experimental results qualitatively, such as the presence of H-bonds between the compound and receptor.  Since they had the experimental results, it would be interesting to see whether there is a correlation between docking scoring (or binding energies) and the IC50 (Table 1). Also, MMPB/SA analysis would be interesting to characterize the binding energies. The authors didn’t report these compounds' docking scoring or binding energies with the two enzymes.

2.      Although molecular docking is a powerful method to investigate the binding event, anonymous compounds could bind to the enzyme's active site. Thus, it needs to be cautious to say that ‘these hybrids may be dual inhibitors of the targeted enzyme, AchE ad BChE (lines 517 and 518). To make such a conclusion, the authors may perform other molecular docking with other enzymes.

3.      It would be interesting to see whether Rivastigmine binds to the active site of enzymes with different binding affinity as a control. The designed compound's binding affinity (scoring) and Rivastigmine must be discussed. Also, Deferiprone, Trolox, and curcumin may be the negative control compounds since the inhibitory activities of the two enzymes were not reported in Table 1. 

Author Response

Response to comments/suggestions of Referee 3 (in red color)

  1. The authors showed that designed compounds bind to the active site of two enzymes. With the docking results, they interpreted the experimental results qualitatively, such as the presence of H-bonds between the compound and receptor. Since they had the experimental results, it would be interesting to see whether there is a correlation between docking scoring (or binding energies) and the IC50 (Table 1). Also, MMPB/SA analysis would be interesting to characterize the binding energies. The authors didn’t report these compounds' docking scoring or binding energies with the two enzymes.

We are thankful to this Referee for the concerns with the MM studies, giving suggestions for the improvement of the quality of the results by enriching the presented data with docking scoring (or binding energies) presentation. We included now Table S1 in the supplementary material, containing the docking scoring for all the compounds and both enzymes. As it can be seen, there is not really a correlation between the docking scoring and the IC50 values determined experimentally, although there is a tendency for an increase of these values with the linker length.

Nevertheless, we are not able to produce binding energies with the GOLD software v. 5.2. In fact, although we do not have Amber package of programs (doi: 10.1002/jcc.20290) for MMPB/SA analysis, we are planning to adopt other docking software in next publications, thus allowing a more complete MD study.

Moreover, by suggestion of another reviewer, redocking of the original ligands was performed again for both enzymes and the authors decided to alter the scoring function from Asp to Goldscore. RMSD values until 1 Å (1.093 for AChE and 0.765 for BChE) were obtained (see experimental part, section 2.3). Figure S3 (in supplementary material) contains the superimposition of the crystallized and docked pose for AChE and BChE. Therefore, docking was performed again for all the compounds and so now, in section 3.2., new figures are presented with examples of docked compounds and the text is rewritten. Some changes in the text of section 3.4. were also done, attending to the new results obtained from MM.

  1. Although molecular docking is a powerful method to investigate the binding event, anonymous compounds could bind to the enzyme's active site. Thus, it needs to be cautious to say that ‘these hybrids may be dual inhibitors of the targeted enzyme, AChE ad BChE (lines 517 and 518). To make such a conclusion, the authors may perform other molecular docking with other enzymes.

The authors agree with the Referee and therefore this sentence was removed from the end of section 3.2.

  1. It would be interesting to see whether Rivastigmine binds to the active site of enzymes with different binding affinity as a control. The designed compound's binding affinity (scoring) and Rivastigmine must be discussed. Also, Deferiprone, Trolox, and curcumin may be the negative control compounds since the inhibitory activities of the two enzymes were not reported in Table 1. 

According to the suggestion of the Referee, the authors performed the docking of the rivastigmine parent drug on the AChE and BChE active sites, in the same conditions of the docking done for the RIV-IND hybrids. The obtained Goldscore scoring function values are included in Table S1 and it can be seen that the obtained values for the drug are lower than those of the studied hybrids. Deferiprone and Trolox (antioxidants) and curcumin (Abeta aggregation inhibitor) were not assayed in vitro for cholinesterase activity and therefore they were not considered in these calculations.

Round 2

Reviewer 2 Report

Comments and Suggestions for Authors

Authors addressed my main concerns

Comments on the Quality of English Language

minor English editing is required

Reviewer 3 Report

Comments and Suggestions for Authors

I am glad to see the improvement of the manuscript. I would like to recommend this manuscript to be published in the Phamaceutics.